# EDGI: Equivariant Diffusion for Planning with Embodied Agents

**Johann Brehmer**[*]
Qualcomm AI Research[†]
jbrehmer@qti.qualcomm.com

**Joey Bose**[* ‡]
Qualcomm AI Research
Mila, McGill University

**Pim de Haan**
Qualcomm AI Research

**Taco Cohen**
Qualcomm AI Research

## Abstract

Embodied agents operate in a structured world, often solving tasks with spatial, temporal, and permutation symmetries. Most algorithms for planning and model-based reinforcement learning (MBRL) do not take this rich geometric structure into account, leading to sample inefficiency and poor generalization. We introduce the Equivariant Diffuser for Generating Interactions (EDGI), an algorithm for MBRL and planning that is equivariant with respect to the product of the spatial symmetry group $SE(3)$, the discrete-time translation group $\mathbb{Z}$, and the object permutation group $S_n$. EDGI follows the Diffuser framework by Janner et al. [2022] in treating both learning a world model and planning in it as a conditional generative modeling problem, training a diffusion model on an offline trajectory dataset. We introduce a new $SE(3) \times \mathbb{Z} \times S_n$-equivariant diffusion model that supports multiple representations. We integrate this model in a planning loop, where conditioning and classifier guidance let us softly break the symmetry for specific tasks as needed. On object manipulation and navigation tasks, EDGI is substantially more sample efficient and generalizes better across the symmetry group than non-equivariant models.

## 1 Introduction

Our world is awash with symmetries. The laws of physics are the same everywhere in space and time—they are symmetric under translations and rotations of spatial coordinates as well as under time shifts.[4] In addition, whenever multiple identical or equivalent objects are labeled with numbers, the system is symmetric with respect to a permutation of the labels. Embodied agents are exposed to this structure, and many common robotic tasks exhibit spatial, temporal, or permutation symmetries. The gaits of a quadruped are independent of whether it is moving East or North, and a robotic gripper would interact with multiple identical objects independently of their labeling. However, most reinforcement learning (RL) and planning algorithms do not take this rich structure into account. While they have achieved remarkable success on well-defined problems after sufficient training, they are often sample-inefficient [Holland et al., 2018] and lack robustness to changes in the environment.

To improve the sample efficiency and robustness of RL algorithms, we believe it is paramount

---

[*]Equal contribution, order determined through a game of table tennis

[†]Qualcomm AI Research is an initiative of Qualcomm Technologies, Inc.

[‡]Work done during an internship at Qualcomm AI Research

[4]This is true in the approximately flat spacetime on Earth, as long as all velocities are much smaller than the speed of light. A machine learning researcher who finds herself close to a black hole may disagree.

37th Conference on Neural Information Processing Systems (NeurIPS 2023).

to develop them with an awareness of their symmetries. Such algorithms should satisfy two key desiderata. First, policy and world models should be equivariant with respect to the relevant symmetry group. Often, for embodied agents this will be a subgroup of the product group of the spatial symmetry group $SE(3)$, the group of discrete time shifts $\mathbb{Z}$, and one or multiple object permutation groups $S_n$. Second, it should be possible to softly break (parts of) the symmetry group to solve concrete tasks. For example, a robotic gripper might be tasked with moving an object to a specific point in space, which breaks the symmetry group $SE(3)$. First works on equivariant RL have demonstrated the potential benefits of this approach [van der Pol et al., 2020, Walters et al., 2020, Mondal et al., 2021, Muglich et al., 2022, Wang and Walters, 2022, Wang et al., 2022, Cetin et al., 2022, Rezaei-Shoshtari et al., 2022, Deac et al., 2023].

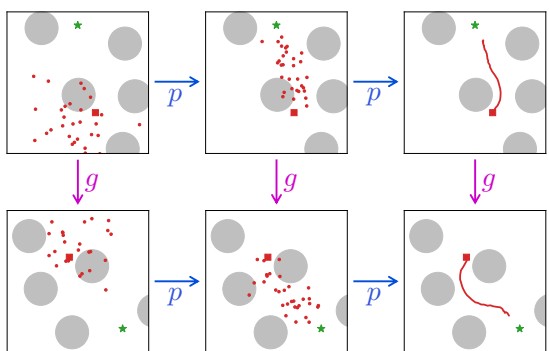

Figure 1: Schematic of EDGI in a navigation task, where the agent (red square) plans the next actions (red dots) to reach the goal (green star) without touching obstacles (grey circles). **Top**: planning as conditional sampling from a diffusion model. **Bottom**: effect of a group action. Equivariance requires the diagram to commute.

However, these works generally only consider small finite symmetry groups such as $C_n$ and do not usually allow for soft symmetry breaking at test time based on a specific task.

In this paper, we introduce the *Equivariant Diffuser for Generating Interactions* (EDGI), an equivariant algorithm for model-based reinforcement learning and planning. EDGI consists of a base component that is equivariant with respect to the full product group $SE(3) \times \mathbb{Z} \times S_n$ and supports the multiple different representations of this group we expect to encounter in embodied environments. Moreover, EDGI allows for a flexible soft breaking of the symmetry at test time depending on the task.

Our work builds on the Diffuser method by Janner et al. [2022], who approach both the learning of a dynamics model and planning within it as a generative modeling problem. The key idea in Diffuser is to train a diffusion model on an offline dataset of state-action trajectories. To plan with this model, one samples from it conditionally on the current state, using classifier guidance to maximize the reward.

Our main contribution is a new diffusion model that is equivariant with respect to the product group $SE(3) \times \mathbb{Z} \times S_n$ of spatial, temporal, and permutation symmetries and supports data consisting of multiple representations. We introduce a new way of embedding multiple input representations into a single internal representation, as well as novel temporal, object, and permutation layers that act on the individual symmetries. When integrated into a planning algorithm, our approach allows for a soft breaking of the symmetry group through test-time task specifications both through conditioning and classifier guidance.

We demonstrate EDGI empirically in 3D navigation and robotic object manipulation environments. We find that EDGI greatly improves the performance in the low-data regime—-matching the performance of the best non-equivariant baseline when using an order of magnitude less training data. In addition, EDGI is significantly more robust to symmetry transformations of the environment, generalizing well to unseen configurations.

## 2  Background

**Equivariant deep learning**. Equivariant networks directly encode the symmetries described by a group $G$ in their architecture. For the purposes of this paper, we are interested in the symmetries of 3D space, which include translations and rotations and are described by the special Euclidean group $SE(3)$, discrete-time translations $\mathbb{Z}$, and object permutations, which are defined using the symmetric group of $n$ elements $\mathbb{S}_n$. We now recall the main definition of equivariance and invariance of functions.

**Definition 1.** A function $f : \mathcal{X} \to \mathcal{Y}$ is called $G$-equivariant if $g \cdot f(x) = f(g \cdot x)$ for all $g \in G$ and $x \in \mathcal{X}$. Here $\mathcal{X}$ and $\mathcal{Y}$ are input and output spaces that carry a $G$ action denoted by $\cdot$. The function $f$ is called $G$-invariant if the group action in $\mathcal{Y}$ is trivial, $g \cdot y = y$.

We will focus on $\mathcal{X} = \mathbb{R}^n$, $\mathcal{Y} = \mathbb{R}^m$, and linear group actions or representations, which are group

homomorphisms $\rho : G \to \mathrm{GL}(\mathbb{R}^k)$. Examples include rotation and permutation matrices. For a more comprehensive introduction to group and representation theory, we direct the interested reader to Appendix A or Esteves [2020] and Bronstein et al. [2021].

For generative modeling, we seek to model $G$-invariant densities. As proven in [Köhler et al., 2020, Bose and Kobyzev, 2021, Papamakarios et al., 2021], given a $G$-invariant prior density it is sufficient to construct a $G$-equivariant map to reach the desired $G$-invariant target density. In Sec. 3, we design $G$-equivariant diffusion architectures to model a distribution of trajectories that are known to be symmetric with respect to the product group $\mathrm{SE}(3) \times \mathbb{Z} \times \mathrm{S}_n$.

**Diffusion models**. Diffusion models [Sohl-Dickstein et al., 2015] are latent variable models that generate data by iteratively inverting a diffusion process. This diffusion process starts from a clean data sample $x \sim q(x_0)$ and progressively injects noise for $i \in [T]$ steps until the distribution is pure noise. The reverse, generative process takes a sample from a noise distribution and denoises it by progressively adding back structure, until we return to a sample that resembles being drawn from the empirical data distribution $p(x)$.

In diffusion models, it is customary to choose a parameter-free diffusion process (e. g. Gaussian noise with fixed variance). Specifically, we may define $q(x_t|x_{t-1})$ as the forward diffusion distribution modeled as a Gaussian centered around the sample at timestep $x_{t-1}$: $q(x_t|x_{t-1}) = \mathcal{N}(x_t; \sqrt{1-\beta_t}x_{t-1}, \beta_t I)$, where $\beta_t$ is a known variance schedule. The reverse generative process is learnable and can be parametrized using another distribution $p_\theta(x_{t-1}|x_t) = \mathcal{N}(x_{t-1}; \mu_\theta(x_t, t), \sigma_t^2 I)$, and the constraint that the terminal marginal at time $T$ is a standard Gaussian—i.e. $p(x_T) = \mathcal{N}(0, I)$. The generative process can be learned by maximizing a variational lower bound on the marginal likelihood. In practice, instead of predicting the mean of the noisy data, it is convenient to predict the noise level $\epsilon_t$ directly [Ho et al., 2020]. Furthermore, to perform low-temperature sampling in diffusion models it is possible to leverage a pre-trained classifier to guide the generation process [Dhariwal and Nichol, 2021]. To do so we can modify the diffusion score by including the gradient of the log-likelihood of the classifier $\bar{\epsilon}_\theta(x_t, t, y) = \epsilon_\theta(x_t, t) - \lambda \sigma_t \nabla_{x_t} \log p(y|x_t)$, where $\lambda$ is the guidance weight and $y$ is the label.

**Trajectory optimization with diffusion**. We are interested in modeling systems that are governed by discrete-time dynamics of a state $s_{h+1} = f(s_h, a_h)$, given the state $s_h$ and action $a_h$ taken at timestep $h$. The goal in trajectory optimization is then to find a sequence of actions $\mathbf{a}_{0:H}^*$ that maximizes an objective (reward) $\mathcal{J}$ which factorizes over per-timestep rewards $r(s_h, a_h)$. Formally, this corresponds to the optimization problem $\mathbf{a}_{0:H}^* = \arg\max_{a_{0:H}} \mathcal{J}(s_0, a_{0:H}) = \arg\max_{a_{0:H}} \sum_{h=0}^{H} r(s_h, a_h)$, where $H$ is the planning horizon and $\tau = (s_0, a_0, \ldots, s_H, a_H)$ denotes the trajectory.

A practical method to solve this optimization problem is to unify the problem of learning a model of the state transition dynamics and the problem of planning with this model into a single generative modeling problem. Janner et al. [2022] propose to train a diffusion model on offline trajectory data consisting of state-action pairs, learning a density $p_\theta(\tau)$. Planning can then be phrased as a conditional sampling problem: finding the distribution $\tilde{p}_\theta(\tau) \propto p_\theta(\tau)c(\tau)$ over trajectories $\tau$ where $c(\tau)$ encodes constraints on the trajectories and specifies the task for instance as a reward function. Diffusion models allow conditioning in a way similar to inpainting in generative image modeling, and test-time reward maximization in analogy to classifier-based guidance.

# 3 Equivariant diffuser for generating interactions (EDGI)

We now describe our EDGI method. We begin by discussing the symmetry group $\mathrm{SE}(3) \times \mathbb{Z} \times \mathrm{S}_n$ and common representations in robotic problems. In Sec. 3.2 we introduce our key novelty, an $\mathrm{SE}(3) \times \mathbb{Z} \times \mathrm{S}_n$-equivariant diffusion model for state-action trajectories $\tau$. We show how we can sample from this model invariantly and break the symmetry in Sec. 3.3. Finally, we discuss how a diffusion model trained on offline trajectory data can be used for planning in Sec. 3.4.

## 3.1 Symmetry and representations

**Symmetry group**. We consider the symmetry group $\mathrm{SE}(3) \times \mathbb{Z} \times \mathrm{S}_n$, which is a product of three distinct groups: 1. the group of spatial translations and rotations $\mathrm{SE}(3)$, 2. the discrete time translation symmetry $\mathbb{Z}$, and 3. the permutation group over $n$ objects $\mathrm{S}_n$. It is important to note, however, that this symmetry group may be partially broken in an environment. For instance, the direction of gravity

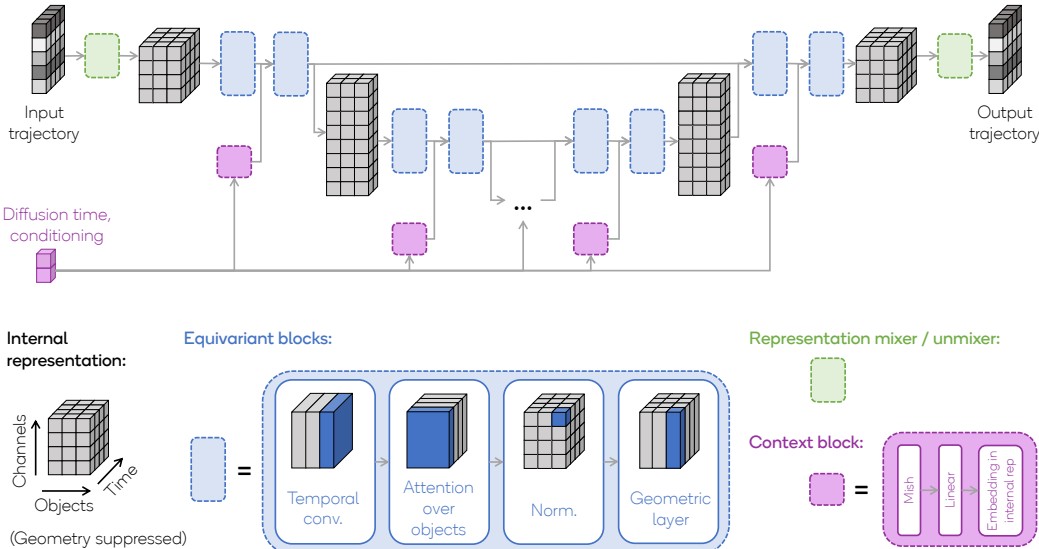

Figure 2: Architecture of our $\mathrm{SE}(3) \times \mathbb{Z} \times \mathrm{S}_n$-equivariant denoising network. Input trajectories (top left), which consist of features in different representations of the symmetry group, are first transformed into a single internal representation (green block). The data are then processed with equivariant blocks (blue), which consist of convolutional layers along the time dimension, attention over objects, normalization layers, and geometric layers, which mix scalar and vector components of the internal representations. These blocks are combined into a U-net architecture. For simplicity, we leave out many details, including residual connections, downsampling, and upsampling layers; see Appendix B.

usually breaks the spatial symmetry group $\mathrm{SE}(3)$ to the smaller group $\mathrm{SE}(2)$, and distinguishable objects in a scene may break permutation invariance. We follow the philosophy of modeling invariance with respect to the larger group and including any symmetry-breaking effects as inputs to the networks.

We require that spatial positions are always expressed relative to a reference point, for example, the robot base or center of mass. This guarantees equivariance with respect to spatial translations: to achieve $\mathrm{SE}(3)$ equivariance, we only need to design an $\mathrm{SO}(3)$-equivariant architecture.

**Data representations**. We consider 3D environments that contain an embodied agent as well as $n$ other objects. We parameterize their degrees of freedom with two $\mathrm{SO}(3)$ representations, namely the scalar representation $\rho_0$ and the vector representation $\rho_1$. Other representations that may occur in a dataset can often be expressed in $\rho_0$ and $\rho_1$. For example, object poses may be expressed as 3D rotations between a global frame and an object frame. We can convert such rotations into two $\rho_1$ representations through the inclusion map $\iota : \mathrm{SO}(3) \hookrightarrow \mathbb{R}^{2 \times 3}$ that chooses the first two columns of the rotation matrix. Such an embedding avoids challenging manifold optimization over $\mathrm{SO}(3)$, while we can still uncover the rotation matrix by orthonormalizing using a Gram-Schmidt process to recover the final column vector. Thus, any $\mathrm{SE}(3)$ pose can be transformed to these two representations.

We assume that all trajectories transform under the regular representation of the time translation group $\mathbb{Z}$ (similar to how images transform under spatial translations). Under $\mathrm{S}_n$, object properties permute, while robot properties or global properties of the state remain invariant. Each feature is thus either in the trivial or the standard representation of $\mathrm{S}_n$.

Overall, we thus expect that data in environments experienced by our embodied agent to be categorized into four representations of the symmetry group $\mathrm{SE}(3) \times \mathbb{Z} \times \mathrm{S}_n$: scalar object properties, vector object properties, scalar robotic degrees of freedom (or other global properties of the system), and vector robotic degrees of freedom (again including other global properties of the system).

### 3.2 Equivariant diffusion model

Our main contribution is a novel $\mathrm{SE}(3) \times \mathbb{Z} \times \mathrm{S}_n$-equivariant diffusion model which leads to an *invariant* distribution over trajectories. Specifically, given an invariant base density with respect to our

chosen symmetry group—an isotropic Gaussian satisfies this property for $\mathrm{SE}(3) \times \mathbb{Z} \times \mathrm{S}_n$—and an denoising model $f$ that is equivariant with respect to the same group, we arrive at a diffusion model that is $\mathrm{SE}(3) \times \mathbb{Z} \times \mathrm{S}_n$-invariant [Köhler et al., 2020, Papamakarios et al., 2021]. Under mild assumptions, such an equivariant map that pushes forward the base density always exists [Bose and Kobyzev, 2021].

We design a novel equivariant architecture for the denoising model $f$. Implemented as a neural network, it maps noisy input trajectories $\tau$ and a diffusion time step $i$ to an estimate $\hat{\epsilon}$ of the noise vector that generated the input. Our architecture does this in three steps. First, the input trajectory consisting of various representations is transformed into an internal representation of the symmetry group. Second, in this representation the data are processed with an equivariant network. Finally, the outputs are transformed from the internal representation into the original data representations present in the trajectory. We illustrate the architecture of our EDGI model in Fig. 2.

**Step 1: Representation mixer**. The input noisy trajectory consists of features in different representations of the symmetry group (see above). While it is possible to mirror these input representations for the hidden states of the neural network, the design of equivariant architectures is substantially simplified if all inputs and outputs transform under a single representation. Hence, we decouple the data representation from the representation used internally for the computation—in a similar fashion to graph neural networks that decouple the data and computation graphs.

**Internal representation**. We define a single internal representation that for each trajectory time step $t \in [H]$, each object $o \in [n]$, each channel $c \in [n_c]$ consists of one[5] $\mathrm{SO}(3)$ scalar $s_{toc}$ and one $\mathrm{SO}(3)$ vector $v_{toc}$. We write $w_{toc} = (s_{toc}, v_{toc}) \in \mathbb{R}^4$. Under spatial rotations $g \in \mathrm{SO}(3)$, these features thus transform as the direct sum of the scalar and vector representations $\rho_0 \oplus \rho_1$:

$$w_{toc} = \begin{pmatrix} s_{toc} \\ v_{toc} \end{pmatrix} \rightarrow w'_{toc} = \begin{pmatrix} \rho_0(g) s_{toc} \\ \rho_1(g) v_{toc} \end{pmatrix} . \tag{1}$$

These internal features transform in the regular representation under time shift and in the standard representation under permutations $\mathbb{P}$ as $w_{toc} \rightarrow w_{to'c} = \sum_o \mathbb{P}_{o'o} w_{toc}$. There are thus no global (not object-specific) properties in our internal representations.

**Transforming input representations into internal representations**. The first layer in our network transforms the input $\tau$, which consists of features in different representations of $\mathrm{SE}(3) \times \mathbb{Z} \times \mathrm{S}_n$, into the internal representation. On the one hand, we pair up $\mathrm{SO}(3)$ scalars and $\mathrm{SO}(3)$ vectors into $\rho_0 \oplus \rho_1$ features. On the other hand, we distribute global features – those a priori unassigned to one of the $n$ objects in the scene – over the $n$ objects.

Concretely, for each object $o \in [n]$, each trajectory step $t \in [H]$, and each channel $c = [n_c]$, we define the input in the internal representation as $w_{toc} \in \mathbb{R}^4$ as follows:

$$w_{toc} = \begin{pmatrix} \sum_{c'} \mathbf{W}^1_{occ'} s_{toc'} \\ \sum_{c'} \mathbf{W}^2_{occ'} v_{toc'} \end{pmatrix} + \begin{pmatrix} \sum_{c'} \mathbf{W}^3_{occ'} s_{t\emptyset c'} \\ \sum_{c'} \mathbf{W}^4_{occ'} v_{t\emptyset c'} \end{pmatrix} . \tag{2}$$

The matrices $\mathbf{W}^{1,2,3,4}$ are learnable and of dimension $n \times n_c \times n_s^{\mathrm{object}}$, $n \times n_c \times n_v^{\mathrm{object}}$, $n \times n_c \times n_s^{\mathrm{global}}$, or $n \times n_c \times n_v^{\mathrm{global}}$, respectively. Here $n_s^{\mathrm{object}}$ is the number of $\mathrm{SO}(3)$ scalar quantities associated with each object in the trajectory, $n_v^{\mathrm{object}}$ is the number of $\mathrm{SO}(3)$ vector quantities associated with each object, $n_s^{\mathrm{global}}$ is the number of scalar quantities associated with the robot or global properties of the system, and $n_v^{\mathrm{global}}$ is the number of vectors of that nature. The number of input channels $n_c$ is a hyperparameter. We initialize the matrices $\mathbf{W}^i$ such that Eq. (2) corresponds to a concatenation of all object-specific and global features along the channel axis at the beginning of training.

**Step 2: $\mathrm{SE}(3) \times \mathbb{Z} \times \mathrm{S}_n$-equivariant U-net**. We then process the data with a $\mathrm{SO}(3) \times \mathbb{Z} \times \mathrm{S}_n$-equivariant denoising network. Its key components are three alternating types of layers. Each type acts on the representation dimension of one of the three symmetry groups while leaving the other two invariant—i. e. they do not mix internal representation types of the other two layers:

- *Temporal layers*: Time-translation-equivariant convolutions along the temporal direction (i. e. along trajectory steps), organized in a U-Net architecture.
- *Object layers*: Permutation-equivariant self-attention layers over the object dimension.

---

[5] Pairing up just one scalar and one vector is a design choice; for systems in which scalar or vectorial quantities play a larger role, it may be beneficial to use multiple copies of either representation here.

- *Geometric layers*: SO(3)-equivariant interaction between the scalar and vector features.

In addition, we use residual connections, a new type of normalization layer that does not break equivariance, and context blocks that process conditioning information and embed it in the internal representation (see Appendix B for more details). These layers are combined into an equivariant block consisting of one instance of each layer, and the equivariant blocks are arranged in a U-net, as depicted in Fig. 2. Between the levels of the U-net, we downsample (upsample) along the trajectory time dimension by factors of two, increasing (decreasing) the number of channels correspondingly.

**Temporal layers**. Temporal layers consist of 1D convolutions along the trajectory time dimension. To preserve SO(3) equivariance, these convolutions do not add any bias and there is no mixing of features associated with different objects nor the four geometric features of the internal SO(3) representation.

**Object layers**. Object layers enable features associated with different objects to interact via an equivariant multi-head self-attention layer. Given inputs $w_{toc}$, the object layer computes

$$\mathbf{K}_{toc} = \sum_{c'} \mathbf{W}^K_{cc'} w_{toc}\,, \mathbf{Q}_{toc} = \sum_{c'} \mathbf{W}^Q_{cc'} w_{toc}\,, \mathbf{V}_{toc} = \sum_{c'} \mathbf{W}^V_{cc'} w_{toc}\,,$$

$$w'_{toc} \propto \sum_{o'} \mathrm{softmax}_{o'}\left(\frac{\mathbf{Q}_{to} \cdot \mathbf{K}_{to'}}{\sqrt{d}}\right) \mathbf{V}_{to'c}, \tag{3}$$

with learnable weight matrices $\mathbf{W}^{K,V,Q}$ and $d$ the dimensionality of the key vector. There is no mixing between features associated with different time steps, nor between the four geometric features of the internal SO(3) representation. Object layers are SO(3)-equivariant, as the attention weights compute invariant SO(3) norms.

**Geometric layers**. Geometric layers enable mixing between the scalar and vector quantities that are combined in the internal representation, but do not mix between different objects or across the time dimension. We construct an expressive equivariant map between scalar and vector inputs and outputs following Villar et al. [2021]: We first separate the inputs into SO(3) scalar and vector components, $w_{toc} = (s_{toc}, v_{toc})^T$. We then construct a complete set of SO(3) invariants by combining the scalars and pairwise inner products between the vectors,

$$S_{to} = \{s_{toc}\}_c \cup \{v_{toc} \cdot v_{toc'}\}_{c,c'}\,. \tag{4}$$

These are then used as inputs to two MLPs $\phi$ and $\psi$, and finally we get output scalars and vectors, $w'_{toc} = (\phi(S_{to})_c, \sum_{c'} \psi(S_{to})_{cc'} v_{toc'})$.

Assuming full expressivity of the MLPs $\phi$ and $\psi$, this approach can approximate any equivariant map between SO(3) scalars and vectors [Villar et al., 2021, Proposition 4]. In this straightforward form, however, it can become prohibitively expensive, as the number of SO(3) invariants $S_{to}$ scales quadratically with the number of channels. In practice, we first linearly map the input vectors into a smaller number of vectors, apply this transformation, and increase the number of channels again with another linear map.

**Step 3: Representation unmixer**. The equivariant network outputs internal representations $w_{toc}$ that are transformed back to data representations using linear maps, in analogy to Eq. (2). Global properties, e. g. robotic degrees of freedom, are aggregated from the object-specific internal representations by taking the elementwise mean across the objects. We find it beneficial to apply an additional geometric layer to these aggregated global features before separating them into the original representations.

**Training**. We train EDGI on offline trajectories without any reward information. We optimize for the simplified variational lower bound $\mathcal{L} = \mathbb{E}_{\tau,i,\epsilon}[\|\epsilon - f(\tau + \epsilon; i)\|^2]$ [Ho et al., 2020]. where $\tau$ are training trajectories, $i \sim \mathrm{Uniform}(0, T)$ is the diffusion time step, and $\epsilon$ is Gaussian noise with variance depending on a prescribed noise schedule.

## 3.3 Invariant sampling and symmetry breaking

We now discuss sampling from EDGI (and, more generally, from equivariant diffusion models). While unconditional samples follow an invariant density, conditional samples may either be invariant or break the symmetry of the diffusion model.

**Invariant sampling**. It is well-known that unconditional sampling from an equivariant denoising model defines an invariant density [Köhler et al., 2020, Bose and Kobyzev, 2021, Papamakarios et al., 2021]. We repeat this result without proof:

**Proposition 1.** Consider a group $G$ that acts on $\mathbb{R}^n$ with representation $\rho$. Let $\pi$ be a $G$-invariant distribution over $\mathbb{R}^n$ and $\epsilon_t : \mathbb{R}^n \to \mathbb{R}^n$ be a $G$-equivariant noise-conditional denoising network. Then the distribution defined by the denoising diffusion process of sampling from $\pi$ and iteratively applying $\epsilon_t$ is $G$-invariant.

We now extend this result to sampling with classifier-based guidance [Dhariwal and Nichol, 2021], a technique for low-temperature sampling based on a classifier $\log p(y|x_t)$ with class labels $y$, or more generally any guide $h(x)$. When the guide is $G$-invariant, guided sampling retains $G$ invariance:

**Proposition 2.** Consider a group $G$ that acts on $\mathbb{R}^n$ with representation $\rho$. Let $\pi$ be a $G$-invariant density over $\mathbb{R}^n$ and $\epsilon_t : \mathbb{R}^n \to \mathbb{R}^n$ be a $G$-equivariant noise-conditional denoising network. Let the guide $h : \mathbb{R}^n \to \mathbb{R}$ be a smooth $G$-invariant function. Further, assume $\rho(g)$ is orthogonal $\forall g \in G$. Define the modified diffusion score $\bar{\epsilon}_\theta(x_t, t, c) = \epsilon_\theta(x_t, t) - \lambda \sigma_t \nabla_{x_t} h(x_t)$ for some guidance weight $\lambda \in \mathbb{R}$. Then the distribution defined by the denoising diffusion process of sampling from $\pi$ and iteratively applying $\bar{\epsilon}$ is $G$-invariant.

*Proof.* The function $h$ has a gradient $\nabla_x h(x)$ that is $G$-equivariant [Papamakarios et al., 2021, Lemma 2]. Thus,

$$
\begin{aligned}
\bar{\epsilon}_\theta(\rho(g)(x_t), t) &= \epsilon_\theta(\rho(g)(x_t), t) - \lambda \sigma_t \nabla_{x_t} h(\rho(g)(x_t)) \\
&= \rho(g)\epsilon_\theta(x_t, t) - \rho(g)\lambda \sigma_t \nabla_{x_t} h(x_t)) \\
&= \rho(g)(\epsilon_\theta(x_t, t) - \lambda \sigma_t \nabla_{x_t} h(x_t)) \, .
\end{aligned}
$$

The modified diffusion score $\bar{\epsilon}$ is therefore $G$-equivariant. Applying Prop. 1, we find that classifier-based guidance samples from a $G$-invariant distribution. $\square$

Proposition 2 applies to $G = \mathrm{SE}(3) \times \mathbb{Z} \times \mathrm{S}_n$ as employed in EDGI, as each group within the product admits an orthogonal matrix representation. Both unconditional sampling from EDGI, as well as sampling guided by a $\mathrm{SE}(3) \times \mathbb{Z} \times \mathrm{S}_n$-invariant classifier (or reward model), is thus $\mathrm{SE}(3) \times \mathbb{Z} \times \mathrm{S}_n$-invariant.

**Symmetry breaking**. However, if the classifier $h(x)$ or $\log p(y|x)$ is *not* $G$-invariant, classifier-guided samples are in general also not $G$-invariant. For example, consider $h(x) = -\|x - x_0\|^2$ for some $x_0 \in \mathbb{R}^n$, which will bias samples to $x_0$. Similarly, conditional sampling on components of $x$ (similar to inpainting) clearly leads to non-invariant samples. As we will argue below, these properties are essential for robotic planning.[6]

### 3.4 Planning with equivariant diffusion

A diffusion model trained on offline trajectory data jointly learns a world model and a policy. Following Janner et al. [2022], we use it to solve planning problems by choosing a sequence of actions to maximize the expected task rewards.

To do this, we use three features of diffusion models. The first is the ability to sample from them by drawing noisy trajectory data from the base distribution and iteratively denoising them with the learned network yielding trajectories similar to those in the training set. For such sampled trajectories to be useful for planning, they need to begin in the current state of the environment. We achieve this by conditioning the sampling process such that the initial state of the generated trajectories matches the current state, in analogy to inpainting. Finally, we can guide this sampling procedure toward solving concrete tasks specified at test time using classifier-based guidance where a regression model is trained offline to map trajectories to task rewards.

**Task-specific symmetry breaking**. By construction, our equivariant diffusion model learns a $\mathrm{SE}(3) \times \mathbb{Z} \times \mathrm{S}_n$-invariant density over trajectories. As shown in the previous section, both unconditional samples (and samples guided by an invariant classifier) reflect this symmetry property—it will be equally likely to sample a trajectory and its rotated or permuted counterpart. However, concrete tasks will often break this invariance, for instance by requiring that a robot or object is brought into a particular location or specifying an ordering over objects to be manipulated in a scene.

---

[6]Task-specific symmetry breaking would be more challenging to implement in classifier-*free* guidance. That would require training a diffusion model that jointly models equivariant unconditional and non-equivariant task-conditional distributions, which will in general be difficult.

| Environment | BCQ | CQL | Standard setting Diffuser | EDGI (ours) | SO(3) generalization Diffuser | EDGI (ours) |
|---|---|---|---|---|---|---|
| Navigation | – | – | $94.9_{\pm 3.9}$ | $95.1_{\pm 3.4}$ | $5.6_{\pm 4.4}$ | $83.3_{\pm 3.5}$ |
| Unconditional | 0.0 | 24.4 | $59.7_{\pm 2.6}$ | $68.7_{\pm 2.5}$ | $38.7_{\pm 2.3}$ | $69.0_{\pm 2.7}$ |
| Conditional | 0.0 | 0.0 | $46.0_{\pm 3.4}$ | $52.0_{\pm 3.6}$ | $16.7_{\pm 2.0}$ | $35.9_{\pm 3.5}$ |
| Rearrangement | 0.0 | 0.0 | $49.2_{\pm 3.3}$ | $47.2_{\pm 3.9}$ | $17.8_{\pm 2.3}$ | $45.0_{\pm 3.6}$ |
| Average | 0.0 | 8.1 | $51.6_{\pm 1.8}$ | $56.0_{\pm 2.0}$ | $24.4_{\pm 1.3}$ | $50.0_{\pm 1.9}$ |

Table 1: Performance on navigation tasks and block stacking problems with a Kuka robot. We report normalized cumulative rewards, showing the mean and standard errors over 100 episodes. Results consistent with the best results within the errors are bold. BCQ and CQL results are taken from Janner et al. [2022]; for Diffuser, we show our reproduction using their codebase. **Left**: Models trained on the standard datasets. **Right**: $SO(3)$ generalization experiments, with training data restricted to specific spatial orientations such that the agent encounters previously unseen states at test time.

As discussed in the previous section, our diffusion-based approach with classifier guidance allows us to elegantly break the symmetry at test time as required. Such a soft symmetry breaking both occurs through conditioning on the current state, by conditioning on a goal state, and through a non-invariant reward model used for guidance during sampling.

# 4 Experiments

We demonstrate the effectiveness of incorporating symmetries as a powerful inductive bias in the Diffuser algorithm with experiments in two environments. The first environment is a 3D navigation task, in which an agent needs to navigate a number of obstacles to reach a goal state. Rewards are awarded based on the distance to the goal at each step, with penalties for collisions with obstacles. The position of the obstacles and the goal state are different in each episode and part of the observation. For simplicity, the actions directly control the acceleration of the agent; both the agent and the obstacles are spherical. Please see Fig. 1 for a schematic representation of this task and Appendix C for more details and the reward structure for this task.

In our remaining experiments, the agent controls a simulated Kuka robotic arm interacting with four blocks on a table. We use a benchmark environment introduced by Janner et al. [2022], which specifies three different tasks: an unconditional block stacking task, a conditional block stacking task where the stacking order is specified, and a rearrangement problem, in which the stacking order has to be changed in a particular way. For both environments, we train on offline trajectory datasets of roughly $10^5$ (navigation) or $10^4$ (manipulation) trajectories. We describe the setup in detail in Appendix D.

**Algorithms**. We train our EDGI on the offline dataset and use conditional sampling to plan the next actions. For the conditional and rearrangement tasks in the Kuka environment, we use classifier guidance following Janner et al. [2022].

As our main baseline, we compare our results to the (non-equivariant) Diffuser model [Janner et al., 2022]. In addition to a straightforward model, we consider a version trained with $SO(3)$ data augmentation. We also compare two model-based RL baselines reported by Janner et al. [2022], BCQ [Fujimoto et al., 2019] and CQL [Kumar et al., 2020]. To study the benefits of the symmetry groups in isolation, we construct two EDGI variations: one is equivariant with respect to $SE(3)$, but not $S_n$; while the other is equivariant with respect to $S_n$, but not $SE(3)$. Both are equivariant to temporal translations, just like EDGI and the baseline Diffuser.

**Task performance**. We report the results on both navigation and object tasks in Tab. 1. For each environment, we evaluate 100 episodes and report the average reward and standard error for each method. In both environments and across all tasks, EDGI performs as well as or better than the Diffuser baseline when using the full training set. In the navigation task, achieving a good performance for the baseline required substantially increasing the model's capacity compared to the hyperparameters used in Janner et al. [2022]. On the Kuka environment, both diffusion-based methods clearly outperform the BCQ and CQL baselines.

**Sample efficiency**. We study EDGI's sample efficiency by training models on subsets of the training

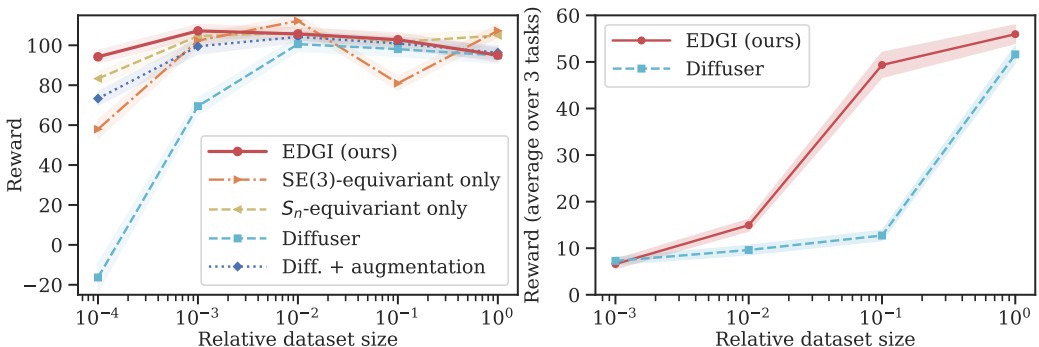

Figure 3: Average reward as a function of training dataset size for EDGI and Diffuser. **Left**: navigation environment. **Right**: Kuka object manipulation, averaged over the three tasks.

data. The results in Fig. 3 show that EDGI achieves just as strong rewards in the Kuka environment when training with only on $10\%$ of the training data, and on the navigation task even when training on only $0.01\%$ if the training data. The Diffuser baseline is much less sample-efficient. Training the Diffuser model with data augmentation partially closes the gap, but EDGI still maintains an edge. Our results provide evidence for the benefits of the inductive bias of equivariant models and matches similar observations in other works for using symmetries in an RL context [van der Pol et al., 2020, Walters et al., 2020, Mondal et al., 2021, Rezaei-Shoshtari et al., 2022, Deac et al., 2023].

**Effects of individual symmetries**. In the left panel of Fig. 3, we also show results for EDGI variations that are only equivariant with respect to $SE(3)$, but not $S_n$, or vice versa. Both partially equivariant methods perform better than the Diffuser baseline, but not as well as the EDGI model equivariant to the full product group $SE(3) \times \mathbb{Z} \times S_n$. This confirms that the more of the symmetry of a problem we take into account in designing an architecture, the bigger the benefits in sample efficiency can be.

**Group generalization**. Finally, we demonstrate that equivariance improves generalization across the $SO(3)$ symmetry group. On both environments, we train EDGI and Diffuser models on restricted offline datasets in which all trajectories are oriented in a particular way. In particular, in the navigation environment, we only use training data that navigates towards a goal location with $x = 0$. In the robotic manipulation tasks, we only use training trajectories where the red block is in a position with $x = 0$ at the beginning of the episode. We test all agents on the original environment, where they encounter goal positions and block configurations unseen during training. We show results for these experiments in Tab. 1. The original Diffuser performs substantially worse, showing its limited capabilities to generalize to the new setting. In contrast, the performance of EDGI is robust to this domain shift, confirming that equivariance helps in generalizing across the symmetry group.

## 5 Related work

**Diffusion-based planning**. The closest work to ours is the original Diffuser paper [Janner et al., 2022], which we used as a baseline. Concurrent to our work, Diffuser was extended by Ajay et al. [2022], who used a separate inverse dynamics model and classifier-free guidance. The key novelty of our work is that we make this approach aware of the symmetry structure of planning problems through a new $SE(3) \times \mathbb{Z} \times S_n$-equivariant denoising network.

**Equivariant deep learning**. Baking in symmetries into deep learning architectures was first studied in the work of Cohen and Welling [2016a] for geometric transformations, and the DeepSet architecture for permutations [Zaheer et al., 2017]. Followup work to group convolutional networks focused on both spherical geometry [Cohen et al., 2018] and building kernels using irreducible group representations [Cohen and Welling, 2016b, Weiler and Cesa, 2019, Cesa et al., 2021]. For symmetries of the 3D space—i. e. subgroups of $E(3)$—a dominant paradigm is to use the message passing framework [Gilmer et al., 2017] along with geometric quantities like positions, velocities, and relative angles [Satorras et al., 2021, Schütt et al., 2021, Batatia et al., 2022].

**Equivariance in RL**. The role of symmetries has also been explored in reinforcement learning problems with a body of work focusing on symmetries of the joint state-action space of an MDP

[van der Pol et al., 2020, Walters et al., 2020, Mondal et al., 2021, Muglich et al., 2022, Wang and Walters, 2022, Wang et al., 2022, Cetin et al., 2022, Rezaei-Shoshtari et al., 2022]. More recently, model-based approaches—like our proposed EDGI—have also benefited from increased data efficiency through the use of symmetries of the environment [Deac et al., 2023]. Concurrently to this work, Brehmer et al. [2023] also experiment with an equivariant Diffuser variation, but their denoising network is based on geometric algebra representations and a transformer architecture.

**Equivariant generative models**. Early efforts in learning invariant densities using generative models utilized the continuous normalizing flow (CNF) framework. A variety of works imbued symmetries by designing equivariant vector fields [Köhler et al., 2020, Rezende and Mohamed, 2015, Bose and Kobyzev, 2021]. As flow-based models enjoy exact density estimation, their application is a natural fit for applications in theoretical physics [Boyda et al., 2020, Kanwar et al., 2020] and modeling equivariant densities on manifolds [Katsman et al., 2021]. Other promising approaches to CNFs include equivariant score matching [De Bortoli et al., 2022] and diffusion models [Hoogeboom et al., 2022, Xu et al., 2022, Igashov et al., 2022]. Our proposed EDGI model extends the latter category to the product group $\mathrm{SE}(3) \times \mathbb{Z} \times \mathrm{S}_n$ and increases flexibility with respect to the data representations.

# 6   Discussion

Embodied agents often solve tasks that are structured through the spatial, temporal, or permutation symmetries of our 3D world. Taking this structure into account in the design of planning algorithms can improve sample efficiency and generalization—notorious weaknesses of RL algorithms.

We introduced EDGI, an equivariant planning algorithm that operates as conditional sampling in a generative model. The main innovation is a new diffusion model that is equivariant with respect to the symmetry group $\mathrm{SE}(3) \times \mathbb{Z} \times \mathrm{S}_n$ of spatial, temporal, and object permutation symmetries. Beyond this concrete architecture, our work presents a general blueprint for the construction of networks that are equivariant with respect to a product group and support multiple representations in the data. Integrating this equivariant diffusion model into a planning algorithm allows us to model an invariant base density, but still solve non-invariant tasks through task-specific soft symmetry breaking. We demonstrated the performance, sample efficiency, and robustness of EDGI on object manipulation and navigation tasks.

While our work shows encouraging results, training and planning are currently expensive. Progress on this issue can come both from more efficient layers in the architecture of the denoising model as well as from switching to recent continuous-time diffusion methods with accelerated sampling.

**Acknowledgements**

We would like to thank Gabriele Cesa, Daniel Dijkman, and Pietro Mazzaglia for helpful discussions.

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

# A Group theory

In this appendix, we provide a basic introduction to Lie groups with an emphasis on the Lie group $SE(3)$ which is used in the main paper.

A Lie group is both a group and a smooth manifold. A manifold is a topological space and thus a Lie group is a topological group that is formally defined below.

**Definition 2.** A topological group is a topological space $G$ equipped with continuous maps that allow group composition $\circ : G \times G \to G$ and group inverses $(\circ)^{-1} : G \to G$.

Now let $g \in G$ be an element of the topological group and consider the map $\bar{g} : G \to G$ given by $\bar{g}(g) = g \circ g'$. If $g \circ g_1 = g \circ g_2$ then $g_1 = g_2$ and thus $\bar{g}$ is injective and for all $g \in G$, and there exists an element—i.e. $g^{-1} \circ g$ such that $\bar{g}(g^{-1} \circ g) = g \circ g^{-1} \circ g' = g'$. Thus, $\bar{g}$ is a bijection.

In a topological space $\bar{g}$ and $\bar{g}^{-1}$ are continuous. Thus, $\bar{g}$ is a homomorphism. This means if $U \subset G$ is open 1.) $g \circ U$ is open 2.) $U \circ g$ is open and inversion is also a homomorphism so $U$ is open if and only if $U^{-1}$ is open. A topological space $\mathcal{X}$ is said to be *homogenous* if $\forall x, y \in \mathcal{X}$ there exists a map $h_{x,y} : \mathcal{X} \to \mathcal{X}$ that is a homomorphism such that $h_{x,y}(x) = y$. Note that a topological group is always homogenous. In words, a homogenous space for a topological group means any group element is reachable by a suitable group homomorphism.

**Matrix Lie groups**. A group of special interest is the group of $n \times n$ invertible matrices with entries in $\mathbb{R}$, $GL_n(\mathbb{R})$. This is a topological group equipped with Euclidean topology. The subgroup $SL_n(\mathbb{R}) \subset GL_n(\mathbb{R})$ is a closed subgroup with the property that $\det = 1$. As an example $SE(3) \subset E(3)$ satisfies this definition and is the group of 3D Euclidean symmetries that include rotations and translations, but not reflections.

A matrix Lie group is a closed subgroup of $GL_n(\mathbb{R})$.[7] This means $SL_n(\mathbb{R})$ is a matrix Lie group as are $O(n)$ and $SO(n)$. Moreover, $O(n)$ and $SO(n)$ are compact. Finally, it is important to note that for matrix Lie groups the exponential and logarithmic maps correspond to the matrix exponential and matrix logarithm. Both of these are infinite power series and have precise connections to the representation theory of matrix Lie groups; namely, the Lie algebra associated with the Lie groups. While we pause further discussion on representation theory here the interested reader is encouraged to read Hall [2013].

$SE(3)$ **as a matrix Lie group**. The group of Euclidean symmetries in 3D is known as the $E(3)$. A closed subgroup of this group is the special Euclidean group, $SE(3)$, and corresponds to rotations and translations in 3D. Commonly, elements of $SE(3)$ can be used to represent rigid body transformations in 3 dimensions. This is because $SE(3) \cong SO(3) \ltimes \mathbb{R}^3$ and thus can be written as,

$$SE(3) = \left\{ \begin{pmatrix} R & x \\ 0 & 1 \end{pmatrix} : R \in SO(3), x \in (\mathbb{R}^3, +) \right\} \tag{5}$$

Represented by this $4 \times 4$ matrix and with the group operation defined by matrix multiplication, this group can be seen as a subgroup of the general linear group $GL_4(\mathbb{R})$.

$SO(3)$ **as a matrix Lie group**. The group of 3D rotations is $SO(3)$. It is a compact matrix Lie group where each element $R \in SO(3)$ is a rotation. To represent $R$ there are various possible choices. The most familiar of them is perhaps $R \in \mathbb{R}^{3 \times 3}$ is as a $3 \times 3$ rotation matrix.

An alternative parameterization is the *rotation vector*. Such a parametrization exploits the Lie algebra of $SO(3)$ which are skew-symmetric matrices $\mathfrak{g}$. Each element of the Lie algebra can be distinctively associated with a vector $\boldsymbol{\omega} \in \mathbb{R}^3$. Given any $\mathbf{v} \in \mathbb{R}^3$, it's related by $\mathfrak{g}\mathbf{v} = \boldsymbol{\omega} \times \mathbf{v}$, with the symbol $\times$ representing the cross product. The length of this vector, symbolized as $\omega = ||\boldsymbol{\omega}||$, represents the rotation angle, while the unit direction, expressed as $e_{\boldsymbol{\omega}} = \frac{\boldsymbol{\omega}}{||\boldsymbol{\omega}||}$, indicates the rotation axis.

Yet another parametrization of rotations can be achieved using *quaternions*. A quaternion, $q$, is often represented as $q = w + xi + yj + zk$ where $w, x, y$, and $z$ are real numbers and $i, j$, and $k$ are mutually orthogonal imaginary units vectors. The unit vectors obey the following rule: $i^2 = j^2 = k^2 = ijk = -1$. A subset of quaternions, known as rotation quaternions, can be used to represent 3D rotations. These specific quaternions have magnitude 1 and can also be thought of as an ordered

---

[7]The field can also be complex $\mathbb{C}$ to allow invertible complex matrices $GL_n(\mathbb{C})$.

pair $(w, v)$ where $w$ is a scalar and $v = (x, y, z)$ is a 3D vector. The scalar part $w$ can be thought of as $\cos(\omega/2)$ and the vector part $v$ as $n \sin(\omega/2)$, where $\omega$ is the angle of rotation and $n$ is the unit vector along the axis of rotation.

Finally, to rotate a point $p$ in 3D space using a quaternion $q$, the point can be represented as a pure quaternion $p' = 0 + xi + yj + zk$. The rotated point $p''$ is then given by: $p'' = q \cdot p' \cdot q^*$. Where $q^*$ is the conjugate of $q$—i.e. if $q = w + xi + yj + zk$, then $q^* = w - xi - yj - zk$.

## B Architecture details

On a high level, EDGI follows Diffuser [Janner et al., 2022]. In the following, we will describe the key difference: our $SE(3) \times \mathbb{Z} \times S_n$-equivariant architecture for the diffusion model.

**Overall architecture**. We illustrate the architecture in After converting the input data in our internal representation (see Sec. 3.2), the data is processed with an equivariant $U$-net with four levels. At each level, we process the hidden state with two residual standard blocks, before downsampling (in the downward pass) or upsampling (in the upward pass).

**Residual standard block**. The main processing unit of our architecture processes the current hidden state with an equivariant block consisting of a temporal layer, an object layer, a normalization layer, and a geometric layer. In parallel, the context information (an embedding of diffusion time and a conditioning mask) is processed with a context block. The hidden state is added to the output of the context block and processes with another equivariant block. Finally, we process the data with a linear attention layer over time. This whole pipeline consists of an equivariant block, a context block, and another equivariant block is residual (the inputs are added to the outputs).

**Temporal layers**. Temporal layers consist of one-dimensional convolutions without bias along the time dimension. We use a kernel size of 5.

**Normalization layers**. We use a simple equivariant normalization layer that for each batch element rescales the entire tensor $w_{toc}$ to unit variance. This is essentially an equivariant version of LayerNorm. The difference is that our normalization layer does not shift the inputs to zero means, as that would break equivariance with respect to $SO(3)$.

**Geometric layers**. In the geometric layers, the input state is split into scalar and vector components. The vector components are linearly transformed to reduce the number of channels to 16. We then construct all $SO(3)$ invariants from these 16 vectors by taking pairwise inner products and concatenating them with the scalar inputs. This set of scalars is processed with two MLPs, each consisting of two hidden layers and ReLU nonlinearities. The MLPs output the scalar outputs and coefficients for a linear map between the vector inputs and the vector outputs, respectively. Finally, there is a residual connection that adds the scalar and vector inputs to the outputs.

**Linear attention over time**. To match the architecture used by Janner et al. [2022] as closely as possible, we follow their choice of adding another residual linear attention over time at the end of each level in the U-net. We make the linear attention mechanism equivariant by computing the attention weights as

**Context blocks**. The embeddings of diffusion time and conditioning information are processed with a Mish nonlinearity and a linear layer, like in Janner et al. [2022]. Finally, we embed them in our internal representation by zero-padding the resulting tensor.

**Upsampling and downsampling**. During the downsampling path, there is a final temporal layer that implements temporal downsampling and increases the number of channels by a factor of two. Conversely, during the upsampling path, we use a temporal layer for temporal upsampling and a reduction of the number of channels.

**Equivariance**. We now demonstrate the equivariance of the EDGI architecture explicitly. For concreteness, we focus on the geometric layers, as they are the most novel, and on both $SE(3)$ transformations and permutations. Similar arguments can be made for the other layers and for equivariance with respect to temporal translations.

Let $w \in \mathbb{R}^{n \times H \times c \times 4}$ be data in our internal representation, such that the entries $w_{toc}$ decompose into $SO(3)$ scalars $s_{toc}$ and $SO(3)$ vectors $v_{toc}$. Let $S(w_{to})$ be the set of all scalars and all pairwise $SO(3)$ inner products between the vectors $v_{to}$, as defined in Eq. (4). The outputs of the geometric layer are

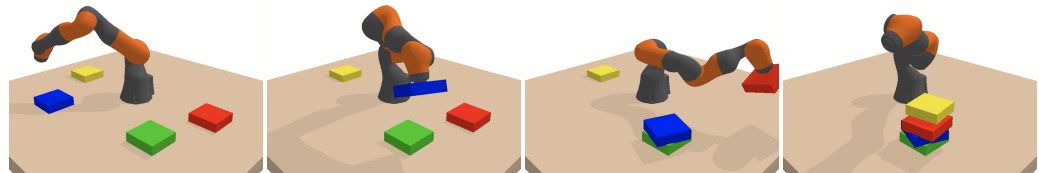

Figure 4: Rendering of the robotic manipulation environment. We show four frames from a single trajectory solving the unconditional block stacking task.

then $f(w)_{toc} = (\phi(S(w_{to})), \sum_{c'} \psi(S(w_{to}))v_{toc'})$.

First, consider what happens under permutations of the objects, $w_{toc} \to w_{t\pi(o)c}$ for a permutation $\pi \in S_n$. We have $f(\pi \cdot w)_{toc} = (\phi(S(w_{t\pi(o)})), \sum_{c'} \psi(S(w_{t\pi(o)}))v_{t\pi(o)c'}) = f(w)_{t\pi(o)c} = (\pi \cdot f(w))_{toc}$. Thus, because this layer "leaves the object dimension untouched", it is equivariant with respect to object permutations.

Next, consider the behavior under spatial transformations. Like most (S)E(3)-equivariant architectures, we deal with translations through canonicalization, defining all coordinates with respect to the center of mass or the robot base, as applicable. This means we only have to analyze the behavior under rotations.

Let $R \in \mathrm{SO}(3)$, such that $R \cdot w = R \cdot (s, v) = (s, R \cdot v)$. By definition, orthogonal matrices leave the inner product invariant, thus $S(R \cdot w) = S(w)$. The geometric layer applied to rotated inputs then gives $f(R \cdot w)_{toc} = (\phi(S(R \cdot w_{to})), \sum_{c'} \psi(S(w_{to}))R \cdot v_{toc'}) = (\phi(S(w_{to})), R \cdot \sum_{c'} \psi(S(w_{to}))v_{toc'}) = (R \cdot f(w))_{toc}$. Hence the geometric layer is equivariant with respect to SE(3).

## C  Navigation experiments

We introduce a new navigation environment. The scene consists of a spherical agent navigating a plane populated with a goal state and $n = 10$ spherical obstacles. At the beginning of every episode, the agent position, agent velocity, obstacle positions, and goal position are initialized randomly (in a rotation-invariant way). We simulate the environment dynamics with PyBullet [Coumans and Bai, 2016–2019].

**Offline dataset**. To obtain expert trajectories, we train a TD3 [Fujimoto et al., 2018] agent in the implementation by Raffin et al. [2021] for $10^7$ steps with default hyperparameters on this environment. We generate $10^5$ trajectories for our offline dataset.

**State**. The state contains the agent position, agent velocity, goal position, and obstacle positions.

**Actions**. The action space is two-dimensional and specifies a force acting on the agent.

**Rewards**. At each time step, the agent receives a reward equal to the negative Euclidean distance to the goal state. In addition, a penalty of $-0.1$ is added to the reward if the agent touches any of the obstacles. Finally, there is an additional control cost equal to $-10^3$ times the force acting on the agent. We affinely normalize the rewards such that a normalized reward of $0$ corresponds to that achieved by a random policy and a normalized reward of $100$ corresponds to the expert policy.

## D  Kuka experiments

We use the object manipulation environments and tasks from Janner et al. [2022], please see that work for details on the environment. In our experiments, we consider three tasks: unconditional stacking, conditional stacking, and block rearrangement. Figure 4 visualizes the unconditional block stacking task. For a fair comparison, we re-implement the Diffuser algorithm while making bug fixes in the codebase of Janner et al. [2022], which mainly included properly resetting the environment.

**State**. We experiment with two parameterizations of the Kuka environment state. For the Diffuser baseline, we use the original 39-dimensional parameterization from Janner et al. [2022].

For our EDGI, we need to parameterize the system in terms of $\mathrm{SE}(3) \times \mathbb{Z} \times \mathrm{S}_n$ representations. We, therefore, describe the robot and block orientations with $\mathrm{SO}(3)$ vectors as follows. Originally, the

robot state is specified through a collection of joint angles. One of these encodes the rotation of the base along the vertical $z$-axis. We choose to represent this angle as a $\rho_1$ vector in the $xy$-plane. In addition, we add the gravity direction (the $z$-axis itself) as another $\rho_1$ vector, which is also the normal direction of the table on which the objects rest. Combined, these vectors define the pose of the base of the robot arm. Rotating gravity direction, and the robot and object pose by $\mathrm{SO}(3)$ can be interpreted as a passive coordinate transformation, or as an active rotation of the entire scene, including gravity. As the laws of physics are invariant to this transformation, this is a valid symmetry of the problem.

The $n$ objects can be translated and rotated. Their pose is thus given by a translation $t \in \mathbb{R}^3$ and rotation in $r \in \mathrm{SO}(3)$ relative to a reference pose. The translation transforms by a global rotation $g \in \mathrm{SO}(3)$ as a vector via representation $\rho_1$. The rotational pose transforms by left multiplication $r \mapsto gr$. The $\mathrm{SO}(3)$ pose is not a Euclidean space, but a non-trivial manifold. Even though diffusion on manifolds is possible De Bortoli et al. [2022], Huang et al. [2022], we simplify the problem by embedding the pose in a Euclidean space. This is done by picking the first two columns of the pose rotation matrix $r \in \mathrm{SO}(3)$. These columns each transform again as a vector with representation $\rho_1$. This forms an equivariant embedding $\iota : \mathrm{SO}(3) \hookrightarrow \mathbb{R}^{2 \times 3}$, whose image is two orthogonal 3-vectors of unit norm. Via the Gram-Schmidt procedure, we can define an equivariant map $\pi : \mathbb{R}^{2 \times 3} \to \mathrm{SO}(3)$ (defined almost everywhere), that is a left inverse to the embedding: $\pi \circ \iota = \mathrm{id}_{\mathrm{SO}(3)}$. Combining with the translation, the roto-translational pose of each object is thus embedded as three $\rho_1$ vectors.

We also tested the performance of the baseline Diffuser method on this reparameterization of the state but found worse results.

**Hyperparameters**. We also follow the choices of Janner et al. [2022], except that we experiment with a linear noise schedule as an alternative to the cosine schedule they use. For each model and each dataset, we train the diffusion model with both noise schedules and report the better of the two results.

