# OpenReview forum: "EDGI: Equivariant Diffusion for Planning with Embodied Agents"
_NeurIPS.cc/2023/Conference — NeurIPS 2023 poster_

### Official Review · Reviewer_kPhe · 2023-06-17

**Soundness:** 2 fair
**Presentation:** 3 good
**Contribution:** 2 fair
**Rating:** 6
**Confidence:** 3

**Summary:**

The paper proposes a new $\mathrm{SE}(3) \times \mathbb{Z} \times \mathrm{S}_n$-equivariant diffusion model based on the symmetriesThe empirical results demonstrate that the proposed EDGI (Equivariant Diffusion for Generating Interactions) model exhibits enhanced efficiency and superior generalization capabilities, even when applied to unseen tasks.  Most RL methods have the issuses of sample-inefficient and lack robustness to the changes of the environment. This paper introduces spatial, temporal, and permutation symmetries into the diffusion model.

**Strengths:**

* The idea is simple and effective. The paper introduces the equivariance into diffusion models, resulting in improvements in generalization performance for unseen tasks.
* The paper designs a novel equivariant U-net architecture that incorporates temporal, object, geometric layers for symmeetries and internal representations for different symmetries.

**Weaknesses:**

* Equavirance has been explored in both RL and model-based approaches. What's the difference with the prior works with equivariance? Need comparison with other baselines that also use equivariance.
* It lacks analysis and ablation studies about the equivariance. For example, there are three types of equivariance. Which has the most improvement to the performance?

**Questions:**

* Needs more analysis and comparison with previous methods that also use equivariance at both method level and experiment level.
* Which has the most improvement to the performance among equivariance?
* Can you explain more clearly how to solve soft symmetry breaking and maybe provide some experimental results?

---

> ### Author Rebuttal · Authors · 2023-08-09
>
> We are very appreciative of the reviewer’s time and effort in reviewing our manuscript. We are grateful to hear that the reviewer finds our idea to be “simple and effective” and that the reviewer thinks the addition of equivariance is indeed reflected in “improved generalization” on unseen tasks. We also pleased to hear that the reviewer found our EDGI architecture to be “novel” in the manner in which we incorporate temporal, object, and SE(3) symmetries.
> We thank the reviewer again for their constructive criticism and now respond to the main clarification points below.
>
>
> **Comparison to other equivariant methods in RL**
>
> We agree that more baselines would be beneficial. We would be grateful for any concrete suggestions of appropriate baselines to compare to.
>
> We are not aware of any RL baselines that are equivariant with respect to similar product groups of all relevant symmetries. To the best of our knowledge, our work is the first equivariant RL method with this property, and as such we believe there is no direct comparison. Similarly, we are not aware of any equivariant methods that solve (MB)RL through generative modeling.
>
> We will however extend our discussion of equivariant RL methods. These involve model-free methods that train SE(3)-equivariant DQN or SAC versions. In contrast, EDGI is a model-based approach that leverages recent advances in conditional generative modeling and is guaranteed to be equivariant to a full product group.
>
> Other recent work has focused on finite subgroups [3] which is an easier modeling problem. We further argue none of the prior work considers equivariance in an offline RL/ conditional generative modeling setting for robotics.
>
> As the reviewer notes, there have been recent developments in equivariant MBRL [4][5]. In [4] the authors consider symmetry with respect to $C_4$, the group of 90-degree rotations, though their evaluation is limited to toy 2D environments. [5] on the other hand uses SE(2)-equivariant steerable convolutions, suitable for image data, but not 3D environments like we consider.
>
> For now, we added one additional baseline: the Diffuser architecture trained with data augmentation. We discuss this experiment and show results in the overall response.
>
>
> **Equivariance ablation studies**
>
> We thank the reviewer for this suggestion. We designed ablation models that are equivariant with respect to only parts of the product symmetry group, and experimented with them on the navigation task.
>
> Our results are shown in Fig. 3 of the rebuttal result page and discussed in the overall response. Essentially, we find that a model that is equivariant with respect to SE(3), but not $S_n$, and a model that is equivariant with respect to $S_n$, but not SE(3), both perform stronger than the Diffuser baseline, but do not quite reach the sample efficiency of EDGI. This provides evidence that both symmetries are important.
>
>
> **Soft symmetry breaking**
>
> We thank the reviewer for pointing out that our explanations of symmetry breaking were not sufficiently clear. We will expand the description in the paper, but the essential idea is the following: Because EDGI builds on the Diffuser paradigm and treats (MB)RL as a diffusion problem, the final behaviour consists of three components – a trained, equivariant diffusion model; a task-specific reward guide; and an initial or current state. The first component is by constructions equivariant, while the latter two allow us to softly break the symmetry *if this is desired*, for instance because of a non-invariant task specification. In the EDGI framework, this form of soft symmetry breaking is very natural.
>
> We already demonstrate this feature in our experiments. In the robotic manipulation environment, we consider both unconditional block stacking, which does not use a reward guide and maintains permutation equivariance, as well as two conditional stacking tasks, in which equivariance is broken by the task specification (e.g. “stack the blue block on top of the red block”). Specifically, each of the conditional and rearrangement tasks in Table 1 requires the Kuka arm to interact with blocks in the environment by stacking them in a particular order (see Fig 1 in our 1-page PDF for a visual of the environment) which requires breaking permutation symmetry due to the specific order of blocks in stack needed to get reward. Please see Fig. 1 in our 1-page PDF for a visual illustration of the task. EDGI achieves good rewards on all tasks, showing the strength of the combination of an equivariant base model and a non-equivariant task-specific reward guide.
>
>
> We thank the reviewer for their valuable feedback and great questions. We hope that our rebuttal fully addresses all the important salient points raised by the reviewer and we kindly ask the reviewer to potentially upgrade their score if the reviewer is satisfied with our responses. We are also more than happy to answer any further questions that arise.
>
>
> **References**
>
> [1] ​​Wang, Dian, Robin Walters, and Robert Platt. "$\mathrm {SO}(2)$-Equivariant Reinforcement Learning." arXiv:2203.04439.
>
> [2] Mondal, Arnab Kumar, et al. "Eqr: Equivariant representations for data-efficient reinforcement learning." ICML 2022.
>
> [3] Zhu, Xupeng, et al. "On Robot Grasp Learning Using Equivariant Models." arXiv:2306.06489.
>
> [4] Deac, Andreea, Théophane Weber, and George Papamakarios. "Equivariant MuZero." arXiv:2302.04798.
>
> [5] Zhao, Linfeng, et al. "Integrating Symmetry into Differentiable Planning with Steerable Convolutions." ICLR 2022.

---

> > ### Comment · Reviewer_kPhe · 2023-08-14
> >
> > Thanks for the response. Most concerns are addressed. I will update the score accordingly.

---

### Official Review · Reviewer_28Jy · 2023-06-25

**Soundness:** 4 excellent
**Presentation:** 3 good
**Contribution:** 3 good
**Rating:** 7
**Confidence:** 3

**Summary:**

This paper proposes an enhancement to a planning/model-based RL method leveraging diffusion models. Specifically, the diffusion model is structured to be equivariant to the known symmetries of reasoning about objects in 3D space, namely translation symmetry, time shift symmetry, and permutation of object labels in the scene. The paper proposes a modeling approach which improves performance on navigation and block stacking benchmarks. The improvement is modest over the baseline Diffuser framework, but taking symmetries into account dramatically improves performance in low data regime and in regimes where the evaluation is performed in a setting that is enforced to be symmetric to the training setting.

**Strengths:**

Very well argumented approach.
Clear writing.
Sound experimental results.

**Weaknesses:**

The paper up to section 3.1 is very repetitive and could be made more concise, leaving more space to introduce some of the modeling details that were left to the appendix.
The ROI of the approach (ratio of improvement over the additional complexity introduced by the model) is limited.


**Questions:**

'training and planning are currently expensive' - do you have more specific details about the overhead, e.g. compared to the Diffuser approach?

**Limitations:**

Limitations of the approach have been adequately discussed, modulo my question above.

---

> ### Author Rebuttal · Authors · 2023-08-09
>
> We thank the reviewer for their positive appraisal of our work! We are delighted to hear that the reviewer finds our approach to have solid motivations and arguments. We are also heartened to hear that the reviewer views our manuscript to be “clearly written” and to contain “sound experimental results”. We now address the main clarification points.
>
>
> **Paper is repetitive up to section 3.1**
>
> We acknowledge the reviewer's concern regarding the lack of concision on our ideas up to section 3.1. We believe it is useful to reiterate the key design objectives in our architecture, especially considering that our work is at the intersection of MBRL, diffusion generative models, and equivariant machine learning. By emphasizing the role of equivariance, we aimed to motivate the key modeling decisions in EDGI. For instance, SE(3) equivariance directly leads to our geometric layer, while permutation symmetry leads to the permutation layer.  Nevertheless, we agree with the reviewer that these points could be more concise. We will update the manuscript to give much more detail on our specific symmetric groups, their corresponding group actions, and representations as well as a primer on group and representation theory in the appendix. We will also add more details on our architecture and explicitly show its equivariance properties, following the logic we sketch in the global  response.
>
>
> **Limited ROI**
>
> We respect the reviewer's apprehension regarding the ROI with respect to the complexity added by EDGI. However, we would like to politely push back as the inclusion of symmetries into the Diffuser architecture is not an optional design decision but rather a key aspect of modeling since the state and action spaces of our environments are *guaranteed* to contain this rich geometric structure. Furthermore, EDGI follows a simple design philosophy and is plug and play in the sense that each layer operates on one specific symmetry group. Please see our global response for a detailed description on EDGI’s equivariant network design including new ablations that highlight the increased sample efficiency of being equivariant to the full product group.
>
>
> **Training and planning complexity**
>
> We appreciate the reviewer's comment regarding the training and planning complexity of EDGI. We provide extensive details on this aspect in our global response as well as the 1-page document which contains training loss curves and inference costs for EDGI as well as the original Diffuser. In summary, we find that EDGI converged about 4x faster during training measured by the number of iterations until we observe a plateau in the loss. Moreover, as we demonstrate in Fig 3. our experiments in section EDGI also enjoys significant gains in sample efficiency over the vanilla Diffuser architecture. However, EDGI does incur a large overhead per iteration. We believe much of this additional overhead can be overcome with implementation optimizations in line with recent efforts [1][2], which we leave as exploration for future work.
>
> We thank the reviewer again for their valuable feedback and great questions. We hope that our rebuttal addresses their questions and concerns and we kindly ask the reviewer to consider upgrading their score if the reviewer is satisfied with our responses. We are also more than happy to answer any further questions that arise.
>
>
> **References**
>
> [1] Liao, Yi-Lun, and Tess Smidt. "Equiformer: Equivariant graph attention transformer for 3d atomistic graphs." arXiv:2206.11990.
>
> [2] Esteves, Carlos, Jean-Jacques Slotine, and Ameesh Makadia. "Scaling Spherical CNNs." arXiv:2306.05420.

---

> > ### Comment · Reviewer_28Jy · 2023-08-10
> >
> > Thank you for the responses.

---

### Official Review · Reviewer_LvgJ · 2023-06-29

**Soundness:** 2 fair
**Presentation:** 3 good
**Contribution:** 3 good
**Rating:** 6
**Confidence:** 4

**Summary:**

The paper introduces the Equivariant Diffuser for Generating Interactions (EDGI), an $SE(3)\times \mathbb{Z} \times S_n$-equivariant diffusion model for model-based reinforcement learning. The proposed method maintains equivariance with spatial symmetry as depicted by $SE(3)$, the discrete time translation symmetry signified by $\mathbb{Z}$, and the object permutation symmetry symbolized by $S_n$. The paper further theoretically analysis on the conditions under which the samples from an equivariant diffusion model will be group invariant or symmetry-breaking. Finally, experimental evaluations were conducted in both manipulation and navigation tasks, demonstrating that the proposed method surpasses the performance of non-equivariant baselines.

**Strengths:**

1. The method considers a large variety of symmetries, all of which are common in many robotic tasks.
2. The concept of using a sequence of three equivariant layers to handle the three distinct symmetries is novel and intriguing.
3. The generalization experiment demonstrates convincing results.

**Weaknesses:**

The experiment section could be more comprehensive.
First, the paper does not provide an ablation study to justify the three symmetries considered. Will removing one or two of the symmetries harms the performance? Which of the three symmetries contributes the most to the success of the architecture?
Second, a data augmentation baseline can also be considered. Though it is widely demonstrated that the equivariant network architecture normally performs better than the learned equivariance through data augmentation, it is still a valuable experiment to validate the proposed network architecture. Moreover, will data augmentation + equivariant network yield even better performance?

**Questions:**

1. Although the proposed sequential approach for handling three different types of symmetries is conceptually sound, a theoretical understanding would be beneficial. Specifically, can the author provide a theoretical analysis that the operation on one of the three equivariant layers will not influence the other two equivariant properties?
2. In some of the experiments (e.g., Navigation in Table 1), EDGI does not significantly outperforms the baseline Diffuser. This is different from what normally is observed in the equivariant learning literature when comparing an equivariant approach vs. a non-equivariant approach. It would be helpful if the authors could provide some analysis on this.
3. Why is the hidden layer in the form of $\rho_0 \oplus \rho_1$? Will adding higher frequency signal in the hidden layer (i.e., $\oplus_0^k \rho_k$ where $k>1$) improve the performance?
4. Some figures of the experimental domains would be helpful (at least in the appendix) to better understand the environments.


**Limitations:**

The paper addresses its limitation, but the discussion could be expanded. For instance, the proposed method seems highly constrained on the input data type, which could make it challenging to extend the proposed method to visual inputs.

---

> ### Author Rebuttal · Authors · 2023-08-09
>
> Thank you for the thorough review and constructive criticism. We are glad that the reviewer thought the way that we incorporate symmetries in EDGI to be “novel and intriguing” and that the reviewer found our generalization experiments to demonstrate “convincing results”. We now address the main questions raised by the reviewer.
>
>
> **Will removing one or two of the symmetries harm the performance?**
>
> Good question. We answer it in detail in our global response to all reviewers. In summary, we implemented a new ablation model  that is equivariant with respect to $S_n$, but not SE(3), and a second ablation model that is equivariant with respect to SE(3), but not to $S_n$. As we show in Fig. 3 of the rebuttal response page, these methods perform much better than the Diffuser baseline, but are not quite as sample-efficient as EDGI with the full symmetry group. This provides more evidence that it is worthwhile to be equivariant with respect to each symmetry found in our robotics environments.
>
>
> **Data augmentation baseline**
>
> Thanks for the suggestion. We trained an additional Diffuser model on the navigation environment, augmenting the data with random SO(3) rotations for each sample. Fig. 3 in the rebuttal result page shows the results. We find that data augmentation substantially improves the Diffuser model in terms of sample efficiency and generalization across the symmetry group, though EDGI still maintains a performance benefit in terms of sample efficiency.
>
>
> **Data augmentation for equivariant networks**
>
> This is an interesting suggestion. However, as long as we only consider data augmentation through symmetry transformations (like rotating the scene or permuting the objects), data augmentations cannot help equivariant methods. Informally, the reason for this is that original and transformed data lie in the same orbit of the symmetry group. This means that both the original and transformed data are acted on by the same network weights in the same way, and lead to the same loss and network updates. Thus we expect no benefit from data augmentation for equivariant networks.
>
>
> **Theoretical analysis of equivariance**
>
> In our global response, we demonstrate explicitly that our novel network layers leave the dimensions of the data it does not act on unaffected, and that it is equivariant with respect to the whole symmetry group.
>
>
> **Performance sometimes on par with baseline Diffuser**
>
> We agree with the reviewer that in our navigation environment when training on a large dataset the performance of EDGI is roughly on par with the baseline. However, we still find two key advantages for our equivariant method: it is much more sample efficient than Diffuser, as shown in Fig. 3 of our paper, and it generalizes more robustly under the symmetry group, as shown in the right column of Table 1 of our paper.
>
>
> **Choice of representations**
>
> We thank the reviewer for this great question. We chose to use  a direct sum of the $\rho_0$ and $\rho_1$ representations for three reasons: it is simple, computations are relatively cheap, and most real-world geometric quantities can be expressed in these representations as they are scalars and vectors.
>
> However, EDGI *can* be extended to include higher representations. Doing so would certainly increase the computational cost of the geometric layers. It is an intriguing question whether that will be offset by benefits in expressivity.  In some domains like molecular dynamics, including higher representations has been very advantageous, see e.g. [1] and the architectures referenced in there. At the same time, there are some results that show that latent scalar and vector representations are enough to represent any equivariant map between vectors [2]. Our EDGI architecture is built on the theoretical foundations of [2] and we leave investigations of higher order representations in the MBRL domain as natural directions for future work.
>
>
> **Illustrations of environments**
>
> Thank you for the suggestion. In Fig. 1 of the rebuttal response page, we show a rendering of the robotic manipulation environment. For the navigation environment, please have a look at Fig. 1 of the paper. We will update our appendix to include these figures.
>
>
> **Constraints on the input data type**
>
> Indeed, we currently assume access to a representation of the data in terms of group representations. We will stress this in the final version of the paper. We consider learning from raw visual inputs and using symmetries in this setting as a natural direction for future work.
>
>
> We thank the reviewer again for their time and efforts reviewing our work. We hope that our rebuttal was successful in addressing all the great points raised by the reviewer and allows the reviewer to consider a fresher evaluation given our rebuttal as context. Finally, please also have a look at our global response. In addition to the important baselines and ablations you suggested, we also show there that EDGI converges faster than the Diffuser baseline.
>
> **References**
>
> [1] Joshi, Chaitanya K., et al. "On the expressive power of geometric graph neural networks." arXiv:2301.09308.
>
> [2] Villar, Soledad, et al. "Scalars are universal: Equivariant machine learning, structured like classical physics." NeurIPS 2021.

---

> > ### Comment · Reviewer_LvgJ · 2023-08-10
> >
> > The reviewer appreciates the author's great rebuttal, most of my concerns are addressed. I would like to increase my evaluation to Weak Accept.

---

### Official Review · Reviewer_FPx8 · 2023-07-09

**Soundness:** 3 good
**Presentation:** 2 fair
**Contribution:** 3 good
**Rating:** 6
**Confidence:** 3

**Summary:**

The paper introduces the Equivariant Diffuser for Generating Interactions (EDGI), a novel algorithm for model-based reinforcement learning (MBRL) and planning. It addresses the challenge of structured environments with spatial, temporal, and permutation symmetries, which are often overlooked by existing planning and MBRL algorithms. EDGI leverages the concept of equivariant diffusion to maintain symmetry under the product of SE(3), Z, and Sn symmetry groups. The algorithm achieves improved sample efficiency and generalization by incorporating a new SE(3) × Z × Sn-equivariant diffusion model that supports multiple representations.

**Strengths:**

1）Novel Approach: The idea of equivariant diffusion is innovative and introduces a fresh perspective on addressing symmetries in planning and MBRL.
2） Multiple Representations: The EDGI algorithm supports multiple representations, which enhances its flexibility and applicability to a wider range of tasks.

**Weaknesses:**

Lack of Clarity in Conceptual Explanation: The introduction of equivariant symmetries could have been more accessible, with clearer explanations of mathematical notations, making it easier for readers to comprehend the concept.
Insufficient Emphasis on Sample Efficiency Improvement: The paper could provide a more explicit explanation of how equivariant symmetries contribute to improved sample efficiency. For instance, the relationship between symmetry breaking and the ability to transfer equivalent trajectories needs further clarification. Does the symmetry breaking approach allow for easy transfer of trajectories between equivalent states, as indicated by Figure 1?
Limited Discussion on Network Design: The paper lacks a thorough discussion on how the network architecture is designed and whether it guarantees the preservation of equivariant symmetries. It would be beneficial to elaborate on the relationship between the network structure and the preservation of equivariant symmetries.

**Questions:**

1) Can you provide a more intuitive explanation of equivariant symmetries and their role in achieving sample efficiency and generalization?
How does the proposed equivariant diffusion model differ from traditional diffusion models, and how does it support multiple representations?
2) Could you elaborate on the specific network design choices and how they ensure the preservation of equivariant symmetries?
3) In practice, what are the computational costs associated with training and using the equivariant diffusion model, particularly for tasks with larger symmetry groups or high-dimensional state and action spaces?
4) Are there any inherent trade-offs between achieving equivariant symmetries and other performance metrics, such as computational efficiency or convergence speed? How does EDGI address these trade-offs, if any?

**Limitations:**

1) Complexity of Symmetry Breaking: Although the paper mentions soft symmetry breaking through conditioning and classifier guidance, it does not delve into the challenges and limitations associated with breaking symmetries in complex environments. Further exploration of the limitations and potential difficulties in achieving effective symmetry breaking would provide a more realistic perspective.

2) Scalability: The scalability of the proposed equivariant diffusion model is not thoroughly discussed. It remains unclear how the algorithm's performance scales with increasing problem complexity or the size of the symmetry group. A deeper investigation into the computational requirements and scalability of the approach would be valuable.

---

> ### Author Rebuttal · Authors · 2023-08-09
>
> We thank the reviewer for their feedback and nuanced comments. We are glad that the reviewer found our contribution to be novel and that it provides “a fresh perspective on symmetries in planning and MBRL”. We also appreciate the fact that the reviewer valued the “flexibility” provided by EDGI through multiple representations allowing it to extend to “a wider range of tasks.” We now address their key questions and concerns.
>
> **Improving accessibility on equivariance**
>
> We thank the reviewer for their constructive criticism regarding our exposition of symmetries and equivariance. We agree with the reviewer that our coverage of the background material may not have been sufficiently clear in the initial manuscript. Toward this end, we will update the paper to include a more comprehensive coverage of the exact groups we consider in this work.
>
> In particular, we will add more detail on the groups SO(3) and $S_n$ along with their representations and actions on our state and action spaces. We will further supplant this with a short primer on group and representation theory in the appendix along with detailed pointers to complete works for the interested reader in the main text.
>
> **Sample efficiency**
>
> We thank the reviewer for highlighting an important aspect, ultimately a main selling point for equivariant architectures: equivariance improves the sample efficiency of training. Essentially, the equivariance constraint ensures that when the network encounters a single sample in training, it learns not only how to transform that sample, but also all data points that can be generated from the sample through symmetry transformations like rotations or permutations. Hence, less samples are necessary to reach a strong, robust performance.
>
> This benefit of equivariance for sample efficiency has been demonstrated both theoretically and empirically in fields from molecular dynamics to robotics, see for instance [1] [2] [3] [4]. Sample efficiency is particularly important in the robotics case, as collecting training data can be expensive.
>
> The reviewer also asks about the role of symmetry breaking in the transfer of trajectories between equivalent states. This is a subtle point (and we will improve its discussion in the final version of the paper). Symmetry breaking consists of two aspects: conditioning on the initial state, as well as on guidance from a reward model. The conditioning on the initial state, together with the equivariance of the denoising model, is what ensures that we can easily transfer learned behaviors to transformed conditions.
>
> The guidance from the reward model, on the other hand, can be used to make the model behave *differently* on rotated situations, if this is desired. For instance, if a task consists of moving an object in a particular direction, then after the initial state is rotated, the agent’s behavior needs to adapt by more than just rotating. Reward guidance allows us to achieve that.
>
> **Network design choices and equivariance**
>
> We value the reviewer's feedback and agree that the paper could be improved by a richer discussion on our network design. How our architecture ensures equivariance is indeed crucial. We address it in the global  response. In a nutshell, we can explicitly show that each layer in EDGI is equivariant to $\mathrm{SE(3)} \times S_n \times \mathbb{Z}$.
>
> **Computational complexity and scaling**
>
> We agree that the computational costs of EDGI is an important aspects that deserve further discussion. We address it in the global response. Essentially, in its current implementation EDGI has substantial overhead over the baseline Diffuser, but we see plenty of room for optimization. What’s more, EDGI converges substantially faster, offsetting the computational overhead. Please see our global response for a more thorough discussion.
>
> As for scalability, EDGI has favorable scaling properties in the three relevant directions – number of objects $n$, number of time steps $H$, and number of channels $c$. Thanks to its modular structure with alternating layers attending to the different data axes, it has a time complexity of $\mathcal{O}(n^2 H c^2)$. When an efficient attention implementation is used, the memory complexity is even just linear in $n$. We will add a thorough discussion of this scaling behaviour to the final version of our paper.
>
> **Symmetry breaking**
>
> We thank the reviewer for pointing out that our explanations of symmetry breaking and its complexities were not sufficiently clear. We will expand the description in the paper, but we believe that the essential idea follows quite naturally from the framework of treating RL as a diffusion problem. The overall behavior of the agent consists of three components – a trained, equivariant diffusion model; a task-specific reward guide; and an initial or current state. The first component is by constructions equivariant, while the latter two allow us to softly break the symmetry *if this is desired*, for instance because of a non-invariant task specification. All this requires is a non-equivariant reward model, for instance a simple MLP.
>
> We would like to thank the reviewer again for their thorough review. We hope we have sufficiently addressed their comments, and we respectfully ask the reviewer to reconsider their impression of the paper and potentially improve the given score. We look forward to discussing any further questions.
>
>
> **References**
>
> [1] Bietti, Alberto, Luca Venturi, and Joan Bruna. "On the sample complexity of learning under geometric stability." NeurIPS 2021.
>
> [2] Behboodi, Arash, Gabriele Cesa, and Taco S. Cohen. "A pac-bayesian generalization bound for equivariant networks." NeurIPS 2022.
>
> [3] Jumper, John, et al. "Highly accurate protein structure prediction with AlphaFold." Nature 2021.
>
> [4] Wang, Dian, et al. "On-robot learning with equivariant models." arXiv:2203.04923.

---

> > ### Comment · Reviewer_FPx8 · 2023-08-18
> > **Thanks for the author's response**
> >
> > The authors' response has addressed my questions. I now have a clearer understanding of the algorithm's details. I am satisfied with this response. I have improved my score.

---

### Author Rebuttal · Authors · 2023-08-09

We would like to thank all reviewers for their thorough reviews and valuable feedback. We are encouraged that they found our approach of equivariant diffusion for planning “innovative” (reviewer **FPx8**), “simple and effective” (**kPhe**), and “very well argumented” (**28Jy**). In particular, they appreciated that our method supported products of multiple symmetry groups (**LvgJ**) and multiple representations (**FPx8**). We are also happy to hear that they found the experimental results “convincing” (**LvgJ**) and the writing “clear” (**28Jy**).


**Showing equivariance (FPx8, LvgJ)**

We now show more thoroughly that our architecture is equivariant.This is best demonstrated explicitly, by proving for each network layer $f(w)$ and each symmetry group $G$ that $f(g \cdot w) = g \cdot f(w)$ for any data $w$ and group elements $g \in G$. Here $\cdot$ denotes the group action.

In the following we will do this for our geometric layers, as they are the most novel, for both SE(3) transformations and permutations.

Let $w \in \mathbb{R}^{n \times H \times c \times 4}$ be data in our internal representation, such that the entries $w_{toc}$ decompose into SO(3) scalars $s_{toc}$ and SO(3) vectors $v_{toc}$. Let $S(w_{to})$ be the set of all scalars and all pairwise $SO(3)$ inner products between the vectors $v_{to}$, as discussed in line 221 in the main paper.

The outputs of the geometric layer are then $f(w)\_{toc} = (\phi( S(w\_{to}) ), \sum_{c’} \psi( S(w\_{to}) ) v\_{toc’} )$.

First, consider what happens under permutations of the objects, $w_{toc} \to w_{t\pi(o)c}$ for a permutation $\pi \in S_n$. We have $f(\pi \cdot w)\_{toc} = (\phi( S(w\_{t\pi(o)}) ), \sum\_{c’} \psi( S(w\_{t\pi(o)}) ) v\_{t\pi(o)c’} ) = f(w)\_{t \pi(o) c} = (\pi \cdot f(w))\_{toc}$. Thus, because this layer “leaves the object dimension untouched”, it is equivariant with respect to object permutations.

An analogous argument can be made for temporal translations.

Finally, consider the behavior under spatial transformations. Like most (S)E(3)-equivariant architectures, we deal with translations through canonicalization, defining all coordinates with respect to the center of mass or the robot base, as applicable. This means we only have to analyze the behavior under rotations.

Let $R \in \mathrm{SO(3)}$, such that $R \cdot w = R \cdot (s, v) = (s, R \cdot v)$. By definition, orthogonal matrices leave the inner product invariant, thus $S(R \cdot w) = S(w)$. The geometric layer applied to rotated inputs then gives $f(R \cdot w)\_{toc} = (\phi( S(R \cdot w\_{to}) ), \sum\_{c’} \psi( S(w\_{to}) ) R \cdot v\_{toc’} ) = (\phi( S(w\_{to}) ), R \cdot \sum\_{c’} \psi( S(w\_{to}) ) v_{toc’} ) = (R \cdot f(w))\_{toc}$. Hence the geometric layer is equivariant with respect to SE(3).


**Computational cost (FPx8, 28Jy)**

There are two aspects to the computational cost of EDGI.
First, EDGI has a more complex computational graph than the baseline Diffuser. In the current implementation, each step (forward plus backward pass) takes roughly 5.5 times as long as that of a baseline model.

We believe that this is not a fundamental property of our method, but just reflects the lack of optimization. We see lots of potential for further speed-ups by optimizing the memory layout of our representations,the contraction paths in tensor multiplications, using optimized attention implementions, or compiling the computational graph. We plan to investigate them in the future.

Second, thanks to its stronger inductive biases, EDGI requires substantially fewer optimizer steps to converge, roughly by a factor of 4. We illustrate this in Fig. 1 of the attached PDF. This at least partially makes up for the computational overhead.


**Additional baselines (kPhe, LvgJ)**

As suggested by reviewer **LvgJ**, we ran the navigation experiment with a version of the Diffuser model trained with data augmentation. We focused on the spatial symmetry group and transformed each sample with SO(3) rotations sampled uniformly from the Haar measure.

The results are shown in Fig. 3 on the attached PDF. We find that data augmentation substantially improves the sample efficiency of the Diffuser model. However, EDGI still maintains a performance benefit in the low-data regimes..


**Ablating the effect of different symmetry groups (LvgJ, kPhe)**

As suggested by the reviewers we investigated the importance of each of the three symmetry group towards final performance. We designed two ablation models: one equivariant with respect to SE(3), but not $S_n$; the other equivariant with respect to $S_n$, but not SE(3). Both models are equivariant to temporal translations, just like EDGI and the baseline Diffuser.

In Fig. 3 on the attached PDF we show how these models perform on the navigation task. Both of these partially equivariant models outperform the baseline Diffuser in terms of sample efficiency, and come closer to EDGI. To our surprise, the permutation-equivariant architecture performed slightly better, indicating large benefits of permutation equivariance. However, using EDGI’s full symmetry group is still the most sample efficient in low-data settings.

We hope that with this global response and the individual responses to the reviewers, we were able to address all questions. We look forward to further discussion.

---

### Decision · Program_Chairs · 2023-09-21

**Decision:**

Accept (poster)

**Comment:**

This paper introduces the equivariant diffuser, a model respecting spatial symmetries to generated interactions in model based LR. It initially received 4 favorable reviews, ratings increased to (7,6,5,6) during the discussion phase. Reviewers generally appreciated the novelty and different perspective this paper brings to the community.

Some weaknesses were raised, linked to clarity / accessibility, limitations of the experiments. The reviewers provided a rebuttal, which answered most of the concerns.

The AC judges that the weaknesses are by far outweighted by the paper strengths, novelty, and recommends acceptance.